# Trends in the Prevalence of Diabetes Mellitus in Pregnancy during the Past Two Decades in Northern Thailand

**DOI:** 10.3390/healthcare11091315

**Published:** 2023-05-04

**Authors:** Phudit Jatavan, Suchaya Luewan, Sirinart Sirilert, Theera Tongsong

**Affiliations:** Department of Obstetrics and Gynecology, Faculty of Medicine, Chiang Mai University, Chiang Mai 50200, Thailand

**Keywords:** diabetes mellitus, gestational diabetes mellitus, pre-gestational diabetes mellitus, pregnancy, trend

## Abstract

***Background:*** The prevalence of gestational diabetes mellitus (GDM) and pre-gestational diabetes mellitus (PDM) has increased dramatically in the past decade in all ethnic groups. The prevalence also varies markedly among different ethnic groups. Each ethnic group must have its own data about GDM/PDM for improvement in women’s health care. We conducted this study with the main objective of assessing recent trends in the prevalence of PDM/GDM among pregnant women in the northern part of Thailand during the past two decades. The secondary objective is to identify the risk factors influencing the prevalence of DM in pregnancies. ***Patients and Methods:*** The maternal–fetal medicine database was accessed to retrieve consecutive obstetric records of women who gave birth in Chiang Mai University Hospital, Thailand, from January 2003 to December 2022. This is a 20-year study period of the same protocol of GDM screening policy, using the 50 g glucose challenge test as a screening test for the average risk group and the 100 g OGTT as a diagnostic test. The women were categorized into GDM, PDM and non-DM groups. Trends or percentage changes in the prevalence of GDM/PDM during the study period were evaluated. Risk factors related to GDM/PDM were identified. ***Results:*** Among 37,027 women who gave birth during the study period, the prevalence of DM in pregnancy was 11.4% (4223 cases), including 214 cases of PDM (0.6%) and 4009 cases of GDM (10.8%). The prevalence of PDM significantly increased from 0.3% in 2003 to 1.5% in 2022; also, the prevalence of GDM significantly increased, dramatically, from 3.4% in 2003 to 22.0% in 2022. The prevalence of GDM increased in recent years in all age groups (adolescent, reproductive and elderly groups), while that of PDM did not significantly change in the adolescent group during the study period. Maternal age and pre-pregnancy BMI significantly increased in the more recent years. Independent factors significantly associated with the prevalence of PDM/GDM include maternal age, pre-pregnancy BMI, higher socio-economic status, and urban areas of residence. Recent time is still an independent risk factor after adjustment for other known factors. ***Conclusions:*** Relatively, GDM and PDM are highly prevalent in the northern part of Thailand, and their prevalence continuously increased during the past two decades. The trend of increased prevalence was evident in all age groups. Increasing maternal age and pre-pregnancy BMI mainly contributed to the increase in the prevalence of GDM and PDM in recent years. Recent time is still an independent risk factor after adjustment for other known factors, indicating that some other unexplained risk factors are associated with the increase in prevalence of DM in recent years, possibly the increase in sedentary lifestyle. Modification of lifestyle, especially reducing pre-pregnancy BMI among reproductive women, may reduce the prevalence of DM in pregnancy.

## 1. Introduction

The prevalence of gestational diabetes mellitus (GDM) has obviously increased in the past decade in several ethnic groups [1]. The prevalence among Asian and Pacific Islander women was reported to be as high as 14.8%, which is higher than that of any other ethnicities, even after adjustment for body mass index and socio-economic status [2]. Additionally, the prevalence also varies with the screening method and criteria used for diagnosis [3]. Lifestyle changes and increased maternal age in the past two decades may have contributed to the increase in prevalence of GDM. According to a large cohort analysis of more than 1.2 million records of women aged 20 years or older [4], elevated pre-pregnancy body mass index (BMI) contributed to GDM in all racial/ethnic groups. The results suggest that decreasing overweight and obesity among women of reproductive age could reduce GDM, associated delivery complications, and future risk of diabetes in the mother and offspring. Apart from the differences in prevalence among populations, the risk of complications of GDM differs significantly among various ethnicities [5,6]. Perinatal outcomes among pregnant women with GDM also vary with race/ethnicity and are associated with socio-cultural differences that may impact glycemic control, genetic variability, the presence of chronic comorbidities, access to antenatal care, as well as quality of antenatal care [6]. Therefore, tailored interventions for GDM management may be required to improve pregnancy outcomes in high-risk ethnic groups. Different ethnic groups may need different strategies of screening, either in terms of method of screening, targeted subgroups of the population or antenatal management suitable for its own population with resource optimization. Each ethnic group in various geographic areas should develop its own data concerning the prevalence of GDM, related obstetric complications as well as later life after GDM pregnancy.

According to our annual reports during the last 30 years, the prevalence of GDM has continuously increased. However, the true prevalence and factors contributing to such an increase has never been thoroughly explored. Therefore, we conducted this study with the main objective of assessing recent trends in the prevalence of pre-gestational diabetes mellitus (PDM) and GDM among pregnant women who gave birth in Maharaj Nakorn Chiang Mai Hospital, in the northern part of Thailand, during the past two decades. The secondary objective is to identify the risk factors influencing the prevalence of DM in pregnancies.

## 2. Patients and Methods

This is a retrospective analytical study using the obstetric database of the maternal–fetal medicine (MFM) unit, department of Obstetrics and Gynecology, Chiang Mai University, Thailand. The study was conducted on pregnant women who gave birth in a single center, Maharaj Nakorn Chiang Mai Hospital, a tertiary hospital and medical teaching school. The study was ethically approved by the institutional review board (The Research Ethics Committee 4; Faculty of Medicine, Chiang Mai University; Study Code: OBG-2565-09130/Research ID: 9130). Our digital obstetric database was first developed by the MFM team in 1990, including baseline characteristics, obstetric and clinical data, as well as pregnancy outcomes, specifically designed and recorded to base obstetric research works and annual reports. However, this study included pregnant women who gave birth from January 2003 to December 2022, during which the policy of GDM screening and diagnosis was well-established with high homogeneity of management. The database was accessed to retrieve all consecutive cases of pregnancies during the study period. The patients were mainly categorized into two groups: pregnant women diagnosed with diabetes mellitus (DM) as the study group and non-DM pregnancies as the control group. The patients in the study group were further categorized into two subgroups: pre-gestational diabetes mellitus (PGM) and gestational diabetes mellitus (GDM). GDM was defined as a condition in which carbohydrate intolerance first occurred or was first detected during pregnancy. During the past two decades, we used the same protocol of screening and diagnosis. Our policy was risk-based screening [3], not universal screening. Pregnancies at high risk of GDM were screened at the first visit to our antenatal care clinic for detecting pre-gestational DM (PGM). The cases of negative screening test were screened again at 24–28 weeks of gestation. Pregnancies at average risk were screened at 24–28 weeks of gestation. The criteria for high risk used in our practice include obesity (BMI > 25), strong family history of type 2 DM (known diabetes in the first degree relatives), previous history of GDM, impaired glucose metabolism, or glucosuria [3], while those for average risk include known history of DM in first-degree relatives, age of greater than 30 years, overweight (BMI of 23–25 kg/m^2^), hypertension or history of cardiovascular disease, history of large-for-date fetuses in a previous pregnancy, and history of unexplained poor obstetric outcomes. The screening method followed the two-step approach, which was based on first screening with the administration of a 50 g oral glucose solution followed by a 1 h glucose determination. Pregnant women whose glucose levels exceeded 140 mg/dL were considered to have a positive screening test and underwent a 100 g, 3 h diagnostic OGTT [3]. GDM was diagnosed in the women who had two or more abnormal values on the 3 h OGTT. Pre-gestational diabetes mellitus (PDM) was defined as DM of any types detected in women prior to the onset of pregnancy.

The main outcomes were the trends of prevalence of GDM and PDM in the past two decades in all women and in specific age groups (adolescent: age of 19 years or less; reproductive age: 20–34 years; and elderly group: 35 years or older).

***Statistical analysis:*** Statistical procedures were performed using statistical package for the social sciences: SPSS; software version 26.0 IBM Corp. Released 2019 (IBM SPSS Statistics for Windows, Version 26.0 Armonk, NY, USA). Descriptive statistics were used to describe demographic data, expressed as mean ± standard deviation. Student’s *t*-test and F-test were used to compare the means of various continuous variables. One-way ANOVA was used as a trend test on time series for continuous variables. Chi-square statistics were used to test for trends in prevalence in yearly periods and to compare the frequency distributions of categorical variables between subgroups, while logistic regression analysis was used to examine the possible risk factors of GDM and PDM. Notably, although some women had more than one pregnancy, the analysis was performed on the number of pregnancies, not the number of women. A *p*-value of less than 0.05 was considered to be of statistical significance.

## 3. Results

Among 37,027 women who gave birth during the study period of 20 years, the mean maternal age increased from 27.7 ± 5.9 years in 2003 to 30.8 ± 5.4 years in 2022 (*p*-value < 0.001), as presented in Figure 1. In addition, the mean pre-pregnancy BMI increased from 22.5 ± 5.4 kg/m^2^ in 2003 to 23.4 ± 5.7 kg/m^2^ in 2022 (*p*-value < 0.001), as presented in Figure 2. The mean ± SD gestational age at first visit was comparable in women with GDM, PDM and controls (14.2 ± 4.2; 16.1 ± 5.4; and 15.3 ± 4.8 weeks; respectively). The mean ± SD gestational age at OGTT was 22.4 ± 8.1 week. Of the women, the prevalence of DM in pregnancy was 11.4% (4223 cases), including 214 cases of PDM (0.6%) and 4009 cases of GDM (10.8%). Trend tests showed that the average annual percentage changes are different from zero, *p*-value <0.001, <0.001 and <0.001 for overall DM, PDM and GDM, respectively. The prevalence of PDM significantly increased from 0.3% in 2003 to 1.5% in 2022 (*p*-value < 0.001), as presented in Figure 3; the prevalence of GDM also significantly increased, dramatically, from 3.4% in 2003 to 22.0% in 2022 (*p*-value < 0.001), as presented in Figure 4.

On analysis of the maternal age subgroups, the increase in the prevalence of DM in pregnancy over time, from 2003 to 2022, was evident in all of the three age groups. The greatest increase in the prevalence of PDM (from 0.2% in 2003 to 2.5%; *p*-value < 0.001) and GDM (from 10.1% in 2003 to 33.5%; *p*-value < 0.001) was demonstrated in the elderly group, as presented in Figure 5 and Table 1. In the adolescent group, the increase in the prevalence of GDM was statistically significant (from 0.4% in 2003 to 8.0%; *p*-value < 0.001), whereas the average prevalence of PDM in the adolescent group was only 0.2% and no significant change occurred over time (*p*-value 0.968). Additionally, the prevalence of PDM and GDM increased significantly with pre-pregnancy BMI. The prevalence of PDM among women with obesity was two times that of women with normal BMI (*p*-value < 0.001). Of note, the prevalence of GDM among women with obesity was 14.4%, as presented in Table 1.

Based on univariate analysis of categorical data, PDM and GDM showed significantly higher prevalence in urban areas (Chiang Mai, presumed to be more urban than the peripheral provinces), recent years of study (the second decade), high parity, higher socio-economic status (mainly based on educational levels), more advanced age group, and higher BMI groups, while multifetal pregnancies tended to increase the prevalence of PDM and GDM, but the increase was not statistically significant. Of note, the prevalence in women attending private clinics was significantly lower than that in the general patients attending our hospital antenatal clinic (*p*-value < 0.001).

On logistic regression analysis, as presented in Table 2, independent factors significantly associated with the increased prevalence of GDM include advanced maternal age, greater pre-pregnancy BMI, recent year of delivery, high parity, urbanity, high socio-economic status and attending antenatal care as a general patient, while multifetal pregnancies tended to increase the prevalence of GDM but not to a statistically significant level. Likewise, the prevalence of PDM is significantly associated with maternal age, pre-pregnancy BMI, year of study, and urbanity, but socio-economic status, parity, number of fetuses and private practice were not significantly associated after adjustment for other risk factors.

## 4. Discussion

New knowledge derived from this study is as follows: (1) The prevalence of DM in pregnancy in our population (northern part of Thailand) is apparently high when compared to that reported in most cohorts in Western countries. (2) PDM and GDM continuously increased during the past two decades, increasing from 0.3% to 1.5% and from 3.4% to 22.0%, respectively. (3) The main reason responsible for the increase in prevalence of DM in pregnancy is likely related to an increase in maternal pre-pregnancy BMI and maternal age in recent years. However, these two factors, which continuously increased in recent years, could not completely explain such an increase in the prevalence of DM. The year of study is still a significant independent risk factor in multivariate analysis. (4) The increase in DM in pregnancy is found in all age groups, but such an increase is more pronounced in elderly gravida. (5) Other factors that might be associated with the development of DM include urbanity of geographical areas, higher socio-economic status and higher parity. However, these factors were not proven to be significantly increased in recent years. Note that the prevalence of DM was significantly lower in private practices. Although the reason is unclear, it is possible that many private clinics prefer to take care of only low-risk pregnancy, referring DM patients for care by the department team.

The overall prevalence of GDM in our population, more than 20% in recent years, is very high, especially when compared to that most commonly quoted in the literature, about 7% [3]. Convincingly, racial factor or ethnicity mainly contributes to such a high rate, since our finding is consistent with those reported in several other previous studies, which demonstrated a higher prevalence of GDM among Asian populations [2,5,6,7]. Actually, the prevalence of both GDM and PDM has increased dramatically in the past decade in all ethnic groups. The prevalence also varies markedly among different ethnic groups [1,5,7,8,9]. Each ethnic group must have its own data about GDM/PDM for improvement in women’s health care.

Interestingly, in this study, the prevalence proportion of PDM compared to that of GDM is rather low, accounting for about 3–4% of DM in pregnancy, which is much different from the figures reported in other previous studies. For example, Correa et al. [10] showed that about 14% of DM in pregnancy represented women with PDM. It is possible that a great number of cases with PDM were not detected prior to the onset of pregnancy since GDM in this study was defined as any pregnant woman in whom abnormal glucose tolerance was first recognized at any time during pregnancy [3]. A more contemporary definition is diabetes diagnosed in the second or third trimester of pregnancy that was not clearly present prior to gestation [11]. Convincingly, with this new definition, not accounting for patients diagnosed in the first trimester, who likely have previously undiagnosed type 2 diabetes, the proportion of PDM must be higher. Accordingly, the relatively low proportion of PDM in this study is likely explained by the different definitions. However, concerning the high prevalence of GDM in our population, it is also possible that the globally used criteria in the diagnosis of GDM might be too sensitive for this group of population, leading to over-diagnosis. Whether each ethnic group should have its own criteria for diagnosis or not is yet to be elucidated. There is a need to comprehensively evaluate whether all cases diagnosed with GDM using traditional criteria are at a higher risk of obstetric complications in different ethnic groups or not. In addition, in comparisons of prevalence of GDM among various populations, the criteria used in screening and diagnosis must be taken into consideration.

We provide evidence that the increase in the prevalence of PDM coincides with that of GDM in the more recent years. The main reasons for the increase are increasing maternal age and maternal BMI, since mean maternal age and BMI significantly increased continuously during the past two decades, which is consistent with the increasing prevalence of DM and is in agreement with previous studies [12,13,14]. Nevertheless, after adjustment for maternal age and BMI on multivariate analysis, the year on the timeline is still an independent risk factor of the increased prevalence, increasing with the more recent years. This finding signifies that there must have been some other unexplained factors contributing to the development of PDM and GDM, which were not included in analysis because of no available data in retrospective review. We hypothesize that enormous changes in lifestyle in the recent years were responsible for an increase in GDM prevalence. In addition to an increase in BMI, modern lifestyle with sedentary jobs or less physical activity secondary to digital working, other underlying disorders such as polycystic ovarian syndrome (PCOS) and metabolic syndrome or pregnancy associated with assisted-reproductive technology might play a role in an increasing rate of DM in pregnancy.

New insights gained from this study may be clinically useful. Although several factors such as advanced maternal age and ethnicity are unlikely or impossible to modify, lifestyle modification such as reducing BMI or overweight/obesity among women of reproductive age is likely to reduce GDM, associated delivery complications, and future risk of DM in the mother and offspring.

The limitations of this study include: (1) Because of its retrospective nature, medical records might not be perfectly reliable, possibly leading to misclassification or misdiagnosis of GDM/PDM. In addition, some important documents were not available for analysis, such as types of PDM, some underlying risk factors of GDM such as polycystic ovarian disease, etc. (2) Obstetric outcomes among pregnancies with GDM/PDM were not included in the analysis. (3) Although the overall sample size was large, the sample size might be too small for some interesting subgroups. For example, the prevalence of DM in multifetal pregnancies tended to increase but was not significantly different from the non-DM controls, which is different from the results in most previous studies. Possibly, this disparity might be due to the low power of the test to show the significance, if it existed. (4) Because risk-based screening was used in this study, not universal screening, the true prevalence might be expectedly higher. However, this is unlikely to affect our main conclusions in terms of trend and associated risk factors.

The strengths of this study include: (1) high homogeneity of the study population, Thai nationality in the northern part of Thailand; the results can represent the prevalence of GDM/PDM in this geographical area. (2) We used the same protocol of screening system and diagnostic method in spite of the long-term study period. (3) The sample size is large enough and the study period is long enough to address the main objective or trends in the prevalence of DM in pregnancy.

## 5. Conclusions

Our findings indicate a relatively high prevalence of GDM and PDM in the northern part of Thailand, which has continuously increased during the past two decades. The trend of increased prevalence was demonstrated in all age groups. Increasing maternal age and pre-pregnancy BMI mainly contributed to the increase in the prevalence of GDM and PDM in recent years. Other unexplained factors, such as modern sedentary lifestyle, might also be responsible for such an increase since, on multivariate analysis, the year of study is still an independent factor associated with the increase, even after adjustment for maternal age and pre-pregnancy BMI. Modification of lifestyle, especially reducing pre-pregnancy BMI among reproductive women, may reduce the prevalence of DM in pregnancy.

## Figures and Tables

**Figure 1 healthcare-11-01315-f001:**
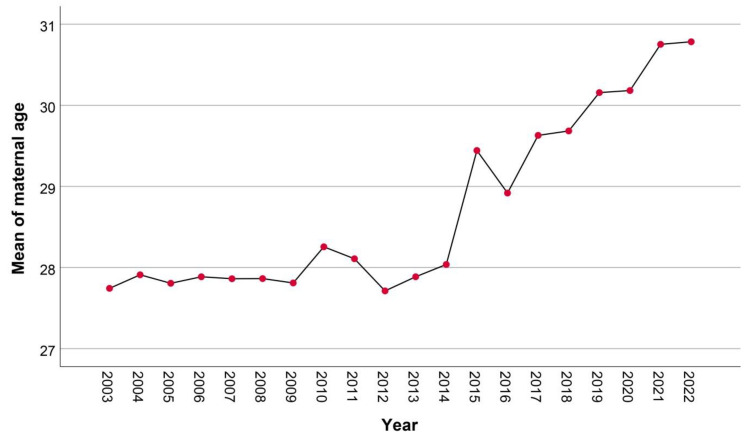
Mean maternal age: trend by years.

**Figure 2 healthcare-11-01315-f002:**
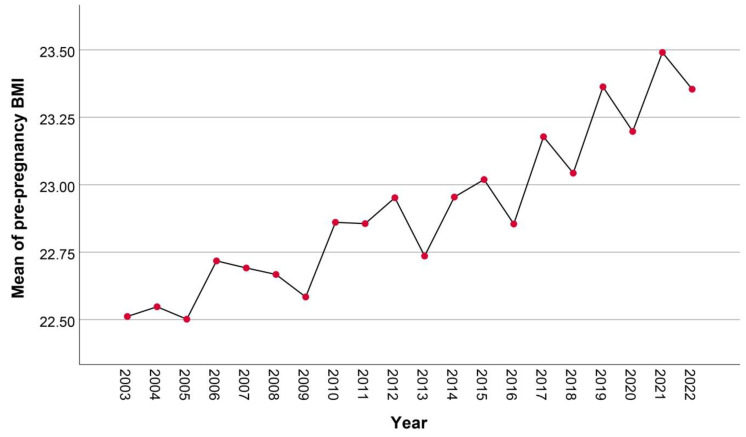
Mean pre-pregnancy BMI: trend by years.

**Figure 3 healthcare-11-01315-f003:**
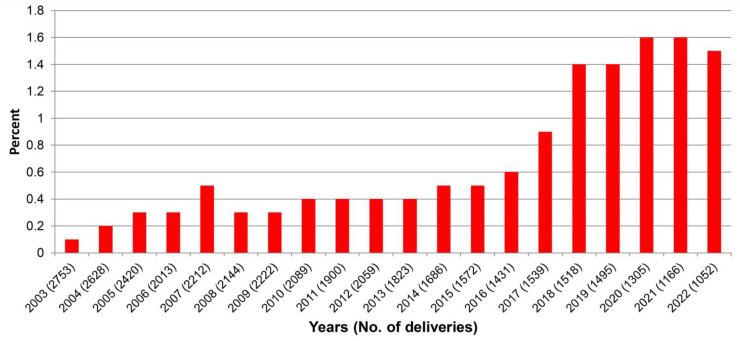
Yearly prevalence of pre-gestational diabetes mellitus (PDM).

**Figure 4 healthcare-11-01315-f004:**
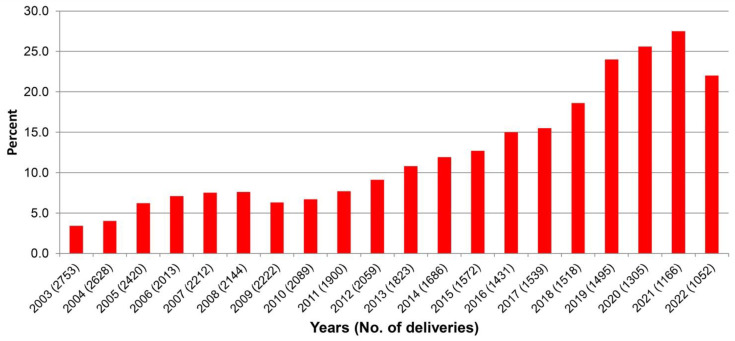
Yearly prevalence of gestational diabetes mellitus (GDM)).

**Figure 5 healthcare-11-01315-f005:**
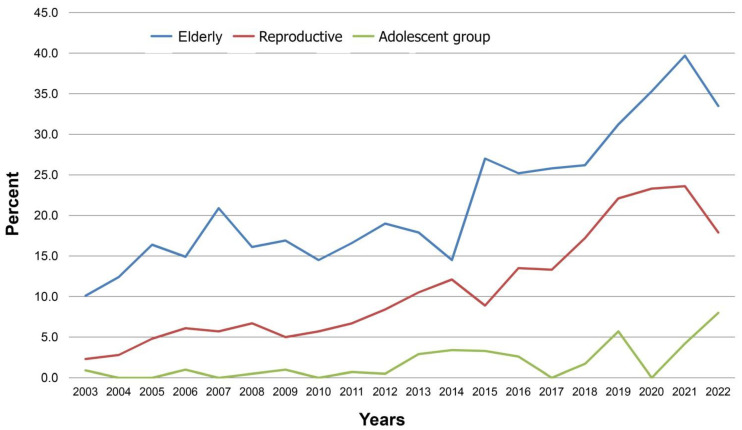
Prevalence of GDM distributions by age groups.

**Table 1 healthcare-11-01315-t001:** Comparisons of the prevalence pre-gestational diabetes mellitus (PDM) and gestational diabetes mellitus (GDM) categorized by risk factors.

Risk Factors	Total *n*	PDM *n* (%)	*p*-Value	GDM *n* (%)	*p*-Value
Residence					
Chiang Mai	25,923	168 (0.6%)	0.007	3030 (11.7%)	<0.001
Others	11,104	46 (0.4%)		979 (8.8%)	
Socio-economic status					
Low	16,410	77 (0.5%)	<0.001	1476 (9.0%)	<0.001
High	16,697	131 (0.8%)		2352 (14.1%)	
Era					
First decade	22,440	69 (0.3%)	<0.001	1433 (6.4%)	<0.001
Second decade	14,587	145 (1.0%)		2576 (17.7%)	
Private practice					
General	32,331	199 (0.6%)	0.018	3744 (11.6%)	<0.001
Private	4545	15 (0.3%)		251 (5.5%)	
Number of fetuses					
Singleton	36,220	208 (0.6%)	0.491	3911 (10.8%)	0.255
Multifetal	787	6 (0.8%)		95 (12.1%)	
Age groups					
Adolescent	2653	4 (0.2%)	<0.001	29 (1.1%)	<0.001
Reproductive age	27,864	144 (0.5%)		2588 (9.3%)	
Elderly	6441	66 (1.0%)		1381 (21.4%)	
BMI groups					
Normal	19,874	88 (0.4%)	<0.001	1769 (8.9%)	<0.001
Overweight	5076	30 (0.6%)		507 (10.0%)	
Obesity	11,792	96 (0.8%)		1694 (14.4%)	
Smoking					
Yes	193	15 (7.8%)	0.157	1 (0.5%)	0.904
No	33,357	3656 (11.0%)		195 (0.6%)	

**Table 2 healthcare-11-01315-t002:** Multivariate analysis for determining the significant risk factors of gestational diabetes mellitus (GDM) and pre-gestational diabetes mellitus (PDM).

	GDM		PDM	
	*p*-Value	Odds Ratio (95% C.I.)	*p*-Value	Odds Ratio (95% C.I.)
Maternal age	<0.001	1.132 (1.124–1.140)	<0.001	1.078 (1.049–1.107)
Pre-pregnancy BMI	<0.001	1.046 (1.039–1.053)	<0.001	1.045 (1.020–1.070)
Year of study	<0.001	1.085 (1.077–1.093)	<0.001	1.104 (1.073–1.137)
Parity	<0.001	1.204 (1.109–1.307)	0.143	1.264 (0.924–1.730)
Residency (Chiang Mai)	<0.001	1.279 (1.173–1.395)	0.048	1.302 (0.925–1.834)
Socio-economic status	<0.001	1.256 (1.156–1.364)	0.240	1.212 (0.879–1.671)
Number of fetus	0.445	1.101 (0.860–1.411)	0.995	1.003 (0.409–2.463)
Private practice	<0.001	0.417 (0.357–0.486)	0.137	0.639 (0.354–1.154)
Smoking	0.126	1.564 (0.881–2.774)	0.933	1.088 (0.151–7.859)

## Data Availability

The datasets analyzed during the current study are available from the corresponding author upon reasonable request.

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
