# Peer review of "Trends in the Prevalence of Diabetes Mellitus in Pregnancy during the Past Two Decades in Northern Thailand"

_healthcare, 2023, doi:10.3390/healthcare11091315_

Round 1

Reviewer 1 Report

Summary: The authors seek to assess recent trends in the prevalence of PDM/GDM among women in their ethnic group, the northern part of Thailand. A 20-year retrospective analytic study involving 37,027 women was conducted and found that the prevalence of the diseases continuously increased. Advanced maternal age, pre-pregnancy BMI and recent time effect mainly contributed to it.

Strengths: (1) The protocol of PDM/GDM screening and diagnosis used during such a long study period remains the same, which makes the conclusions more solid. (2) Different risk factors of PDM/GDM were studied, including general or private practice.

Areas of improvement:

1. The contents of Part2 "Patients and Methods" should be supplemented and clarified for some details, especially from line 96-112:

(1) Could you provide any reference for the risk-based screening policy used in this study?

(2) Please provide reference or specific definitions to "strong family history of type 2DM", "impaired glucose metabolism" and "the 100g 3-hour diagnostic OGTT".

(3) The definition of PDM is confusing in line 111. Does PDM include women who reached T2DM diagnostic criteria for the first time at their first visit to the antenatal care clinic in this study?

2. Is there any chance that the same woman gave birth more than once in the past 20 years? Their parity, advanced age and GDM history should be considered and the sample size could be affected.

3. The population who received assisted reproduction technology to get pregnant is increasing. Previous studies demonstrated that women who went through IVF had a higher risk of GDM, possibly due to PCOS or other pregestational glucose metabolism disorders. If possible, any data concerning IVF could be presented to further analyze the possible reasons for the increasing prevalence of PDM/GDM.

4. As for the Results part, the average gestational week of their first visit to the antenatal clinic as well as OGTT should be provided.

5. If possible, other pregnancy outcomes including complications or birth outcomes could be provided in this study to enrich the contents, although it was stated as a limitation of the study.

6. The first paragraph of Discussion should be extended. The "other factors" and "unexplained reasons" mentioned in the Abstract should be further discussed.

7. It is interesting that the prevalence of the disease was significantly lower in private clinical practices. Any possible reasons can be explained for this result? Is there any difference in medical resources such as nutritional consultations between general practices and private practices in Thailand?

8. The prevalence of PDM was relatively low in this study. It is suggested that the data on T2DM incidence in the general population could be provided to make a comparison.

9. Could you provide the number of annual delivery of the study centre?

10. The cited references in this study should be up-to-date, which can be improved greatly, especially in the Introduction part.

Author Response

Reviewer: 1 (highlighted in blue)

Comments and Suggestions for Authors

Summary: The authors seek to assess recent trends in the prevalence of PDM/GDM among women in their ethnic group, the northern part of Thailand. A 20-year retrospective analytic study involving 37,027 women was conducted and found that the prevalence of the diseases continuously increased. Advanced maternal age, pre-pregnancy BMI and recent time effect mainly contributed to it.

Strengths: (1) The protocol of PDM/GDM screening and diagnosis used during such a long study period remains the same, which makes the conclusions more solid. (2) Different risk factors of PDM/GDM were studied, including general or private practice.

Areas of improvement:

  1. The contents of Part2 "Patients and Methods" should be supplemented and clarified for some details, especially from line 96-112:

 (1) Could you provide any reference for the risk-based screening policy used in this study?

Response: The references is provided.

 (2) Please provide reference or specific definitions to "strong family history of type 2DM", "impaired glucose metabolism" and "the 100g 3-hour diagnostic OGTT".

Response: The references / more specific definitions are provided.

(3) The definition of PDM is confusing in line 111. Does PDM include women who reached T2DM diagnostic criteria for the first time at their first visit to the antenatal care clinic in this study?

Response: Yes, GDM was based on the first diagnosis during pregnancy while PDM was included only the cases documented prior to pregnancy. In revised MS, the definition of PDM is more clearly described.

  1. Is there any chance that the same woman gave birth more than once in the past 20 years? Their parity, advanced age and GDM history should be considered and the sample size could be affected.

Response: Yes, there is the chance. Some cases in both GDM and non-DM (controls) giving birth more than once in this study. The analysis was performed on the number of pregnancies not the number of women. This is clearly indicated in part of “Statitical analysis” in “Patients and Methods”

  1. The population who received assisted reproduction technology to get pregnant is increasing. Previous studies demonstrated that women who went through IVF had a higher risk of GDM, possibly due to PCOS or other pregestational glucose metabolism disorders. If possible, any data concerning IVF could be presented to further analyze the possible reasons for the increasing prevalence of PDM/GDM.

Response: Unfortunately, we cannot provide this data because the two factors were not targetedly extracted from medical records during the review. Anyway, we add this issue as a limitation of this study; included in the paragraph of “weakness” in “Discussion”.

  1. As for the Results part, the average gestational week of their first visit to the antenatal clinic as well as OGTT should be provided.

Response: Average weeks of the first visits and weeks for OGTT was added in the first paragraph of “Results”, as highlighted.

  1. If possible, other pregnancy outcomes including complications or birth outcomes could be provided in this study to enrich the contents, although it was stated as a limitation of the study.

Response: We apologize that we do not present the pregnancy outcomes in this study, since the objective of this study aimed to focus on epidemiology, specifically trends in prevalence of DM. In fact, we published the pregnancy outcomes among pregnancy with GDM among our pregnant women in year 2015 (reference below). Reanalysis is expected to be much different. Accordingly, we make kindly request not to include pregnancy outcomes.

Srichumchit S, Luewan S, Tongsong T. Outcomes of pregnancy with gestational diabetes mellitus. Int J Gynaecol Obstet 2015 Dec;131(3):251-4. doi: 10.1016/j.ijgo.2015.05.033.

  1. The first paragraph of Discussion should be extended. The "other factors" and "unexplained reasons" mentioned in the Abstract should be further discussed.

Response: The other factors are now elaborated in the 4th paragraph of “Discussion”, as follows: “This finding signifies that there must have been some other unexplained factors contributing to the development of PDM and GDM, which were not included in analysis because of no available data in retrospective review. We hypothesize that enormous changes in life style in the recent years were responsible for an increase in GDM prevalence. In addition to an increase in BMI, modern life style with sedentary jobs or less physical activity secondary to digital working, other underlying disorders like polycystic ovarian syndrome (PCOS) and metabolic syndrome or pregnancy associated with assisted-reproductive technology might play a role in an increasing rate of DM in pregnancy.”

  1. It is interesting that the prevalence of the disease was significantly lower in private clinical practices. Any possible reasons can be explained for this result? Is there any difference in medical resources such as nutritional consultations between general practices and private practices in Thailand?

Response: The possible reason is provided at the end of 1st paragraph of “Discussion”

  1. The prevalence of PDM was relatively low in this study. It is suggested that the data on T2DM incidence in the general population could be provided to make a comparison.

Response: We do not mean that PDM is lower during pregnancy when compared to DM in non-pregnant women but the prevalence proportion of PDM is rather low when compared to GDM, Different from some other studies. (The reference is also provided). This is elaborated in the 3rd paragraph, as highlighted.

  1. Could you provide the number of annual delivery of the study centre?

Response: The total number of annual deliveries is now added in X-axis of Figure 3 and 4 (the number of women continuously decreased, while the rate of DM increases.)

  1. The cited references in this study should be up-to-date, which can be improved greatly, especially in the Introduction part.

Response: The references have been updated both in “Introduction” and “Discussion”.

Reviewer 2 Report

Dear authors,

Thank you for this interesting study.
I have several comments about this study.

Firstly, you talk in this article about PDM and GDM. It is obviously interesting to have some uploaded data about these 2 prevalences, because of the increase you've noticed in the recent years. However, in the PDM group, I would have detailed the different types of diabetes. As you know, T1D and T2D or MODY diabetes are very differents in term of pathophysiology, and complications. It would be interesting to know if only T2D has increased in the past recent years, because of the increase of BMI and sedentary lifestyle, or if T1D has also increased in the same period of time, because it is not likely to have increased because of the same reasons.
Moreover, PDM in your study could have been underdiagnosed, because every glycemic disorder found during the pregnancy was considered as GDM, as far as I understand. But some pre-gestational glycemic disorders could not have been diagnosed before the pregnancy, as symptoms are often scarce, in T2D especially. And so these disorders, diagnosed during pregnancy, could be misclassified as GDM, which could then be considered higher than in other studies.

Then, you say that GDM is more frequent in your studies than in others. It may be because of the way you diagnose GDM. The protocol for GDM diagnosis is not the same all around the world. For example in France, for women who have risk factors only, we use fasting glucose level >/=0.92g/L during the first trimester of the pregnancy and, if normal, 75g 2h-OGTT between 24 and 28 weeks of pregnancy.
So it seems to be difficult to compare prevalences of something that is not diagnosed the same way everywhere. And that could also explain why your prevalence is different from others.
The year of diagnosis is also important. As you show in your study, prevalence is increasing especially in the past recent years. Prevalence that has been determined 5 or 10 years ago could so be lower than today. You should consider only recent studies for the prevalence of GDM to compare with your work.

Finally, you mention that the increase of GDM and PDM could be explained by the increase of BMI and maternal age but not only. Some confusion factors could have interfere with the result, and your hypothesis is that sedentary lifestyle could be one of those factors.
I would rather say that others factors are involved, because sedentary lifestyle is accompanied by the increase of BMI, and so would may not be an independant factor for me. It could be factors like pollution, I imagine, as the increase of PDM and GDM concerns especially urban areas.
Another factor that has not been taken into consideration in your study is smoking. Smoking is a known risk factor for the occurrence of T2D and also GDM in women (Zhang C, Tobias DK, Chavarro JE, Bao W, Wang D, Ley SH, Hu FB. Adherence to healthy lifestyle and risk of gestational diabetes mellitus: prospective cohort study. BMJ. 2014 Sep 30;349:g5450. doi: 10.1136/bmj.g5450. PMID: 25269649; PMCID: PMC4180295.) (Pan A, Wang Y, Talaei M, Hu FB, Wu T. Relation of active, passive, and quitting smoking with incident type 2 diabetes: a systematic review and meta-analysis. Lancet Diabetes Endocrinol. 2015 Dec;3(12):958-67. doi: 10.1016/S2213-8587(15)00316-2. Epub 2015 Sep 18. PMID: 26388413; PMCID: PMC4656094.).
And, despite the decreasing prevalence of smoking  over the recent time, it is not always the case for women who are likely to smoke more than before, especially in some age groups and ethnics (Zhang G, Zhan J, Fu H. Trends in Smoking Prevalence and Intensity between 2010 and 2018: Implications for Tobacco Control in China. Int J Environ Res Public Health. 2022 Jan 7;19(2):670. doi: 10.3390/ijerph19020670. PMID: 35055491; PMCID: PMC8776183.).
It would be interesting to add this data in your study, if possible, as it is a confounding factor for the prevalence of PDM and GDM.

Author Response

Reviewer 2 (highlighted in red)

Comments and Suggestions for Authors

Dear authors,

Thank you for this interesting study.

I have several comments about this study.

Firstly, you talk in this article about PDM and GDM. It is obviously interesting to have some uploaded data about these 2 prevalences, because of the increase you've noticed in the recent years. However, in the PDM group, I would have detailed the different types of diabetes. As you know, T1D and T2D or MODY diabetes are very differents in term of pathophysiology, and complications. It would be interesting to know if only T2D has increased in the past recent years, because of the increase of BMI and sedentary lifestyle, or if T1D has also increased in the same period of time, because it is not likely to have increased because of the same reasons.

Response: We apologize that we can not provide the data of DM types because, on medical record review, the data of types of DM were not specifically extracted for analysis. We add comment on this issue as a limitation, as highlighted in part “limiations” of “Discussion”

Moreover, PDM in your study could have been underdiagnosed, because every glycemic disorder found during the pregnancy was considered as GDM, as far as I understand. But some pre-gestational glycemic disorders could not have been diagnosed before the pregnancy, as symptoms are often scarce, in T2D especially. And so these disorders, diagnosed during pregnancy, could be misclassified as GDM, which could then be considered higher than in other studies.

Response: We absolutely agree with the reviewers, because we used the traditional criteria of first diagnosed during pregnancy. We add comment on this issue in “Discussion”, as highlighted in red in the 3rd paragraph of “Discussion”.

Then, you say that GDM is more frequent in your studies than in others. It may be because of the way you diagnose GDM. The protocol for GDM diagnosis is not the same all around the world. For example in France, for women who have risk factors only, we use fasting glucose level >/=0.92g/L during the first trimester of the pregnancy and, if normal, 75g 2h-OGTT between 24 and 28 weeks of pregnancy.

So it seems to be difficult to compare prevalences of something that is not diagnosed the same way everywhere. And that could also explain why your prevalence is different from others.

Response: We add the comment on this problem at the end of the 3rd paragraph of “Discussion” as highlighted in red.

The year of diagnosis is also important. As you show in your study, prevalence is increasing especially in the past recent years. Prevalence that has been determined 5 or 10 years ago could so be lower than today. You should consider only recent studies for the prevalence of GDM to compare with your work.

Response: In the revised MS, references for comparisons are updated.

Finally, you mention that the increase of GDM and PDM could be explained by the increase of BMI and maternal age but not only. Some confusion factors could have interfere with the result, and your hypothesis is that sedentary lifestyle could be one of those factors.

I would rather say that others factors are involved, because sedentary lifestyle is accompanied by the increase of BMI, and so would may not be an independant factor for me. It could be factors like pollution, I imagine, as the increase of PDM and GDM concerns especially urban

areas.

Response: Thank you for the comment. We agree that sedentary life might not only be associated with an increased BMI but also other subtle risk factor even not increase in BMI. We add the comment on other factors at the end of the 4th paragraph of “Discussion” as highlighted in blue (the same response to the reviewer 1)

Another factor that has not been taken into consideration in your study is smoking. Smoking is a known risk factor for the occurrence of T2D and also GDM in women (Zhang C, Tobias DK, Chavarro JE, Bao W, Wang D, Ley SH, Hu FB. Adherence to healthy lifestyle and risk of gestational diabetes mellitus: prospective cohort study. BMJ. 2014 Sep 30;349:g5450. doi: 10.1136/bmj.g5450. PMID: 25269649; PMCID: PMC4180295.) (Pan A, Wang Y, Talaei M, Hu FB, Wu T. Relation of active, passive, and quitting smoking with incident type 2 diabetes: a systematic review and meta-analysis. Lancet Diabetes Endocrinol. 2015 Dec;3(12):958-67. doi: 10.1016/S2213-8587(15)00316-2. Epub 2015 Sep 18. PMID: 26388413; PMCID: PMC4656094.).

Response: Thank you for the comment. Smoking is very rare among women in our population and pregnant women (only 0.5%). However, in revised MS, we additionally analyze on the effects of smoking, as presented in Table 1 and 2. The analysis causes all figures in Table 2 minimally changes.

And, despite the decreasing prevalence of smoking over the recent time, it is not always the case for women who are likely to smoke more than before, especially in some age groups and ethnics (Zhang G, Zhan J, Fu H. Trends in Smoking Prevalence and Intensity between 2010 and 2018: Implications for Tobacco Control in China. Int J Environ Res Public Health. 2022 Jan 7;19(2):670. doi: 10.3390/ijerph19020670. PMID: 35055491; PMCID: PMC8776183.).

It would be interesting to add this data in your study, if possible, as it is a confounding factor for the prevalence of PDM and GDM.

Response: In revised MS, we add smoking data.
